# Relationship between Training Factors and Injuries in Stand-Up Paddleboarding Athletes

**DOI:** 10.3390/ijerph18030880

**Published:** 2021-01-20

**Authors:** Arkaitz Castañeda-Babarro, Julio Calleja-González, Aitor Viribay, Diego Fernández-Lázaro, Patxi León-Guereño, Juan Mielgo-Ayuso

**Affiliations:** 1Health, Physical Activity and Sports Science Laboratory, Department of Physical Activity and Sports, Faculty of Psychology and Education, University of Deusto, 48007 Bizkaia, Spain; arkaitz.castaneda@deusto.es (A.C.-B.); patxi.leon@deusto.es (P.L.-G.); 2Physiotherapy Department, Institute of Biomedicine (IBIOMED), Campus of Vegazana, University of Leon, 24071 Leon, Spain; 3Department of Physical Education and Sport, Faculty of Education and Sport, University of the Basque Country (UPV/EHU), 01007 Vitoria, Spain; julio.calleja.gonzalez@gmail.com; 4Glut4Science, Physiology, Nutrition and Sport, 01004 Vitoria-Gasteiz, Spain; aitor@glut4science.com; 5Department of Cellular Biology, Histology and Pharmacology, Faculty of Health Sciences, Campus of Soria, University of Valladolid, 42003 Soria, Spain; diego.fernandez.lazaro@uva.es; 6Neurobiology Research Group, Faculty of Medicine, University of Valladolid, 47005 Valladolid, Spain; 7Department of Health Sciences, Faculty of Health Sciences, University of Burgos, 09001 Burgos, Spain; 8ImFINE Research Group, Department of Health and Human Performance, Faculty of Physical Activity and Sport Sciences-INEF, Polytechnic University of Madrid, 28040 Madrid, Spain

**Keywords:** stand-up paddleboarding, SUP, injury, prevention, resistance training

## Abstract

Stand-up paddleboarding (SUP) is an increasingly popular sport but, as in other sports, there is an injury ratio associated with practicing it. In other types of sport, some factors have been linked to the likelihood of suffering an injury, among which stretching, core training and resistance training may be considered the most significant. Therefore, the main aim of this study was to identify the training factors that could influence injuries suffered by participants in international SUP competitions. Ninety-seven questionnaires were collected from paddlers who participated in an international SUP circuit, with epidemiological data being gathered about injuries and different questions related to the training undertaken. A multi-factor ANOVA test was used to identify the factors which influence the state of injury. Results showed that almost 60% of injuries occurred in the arms or in the upper thoracic region, around 65% of which were in tendons or muscles and, in almost half of cases, were related to overuse. Likewise, the results showed that athletes with injury performed fewer resistance training sessions per week (*p* = 0.028), over fewer months per year (*p* = 0.001), more weekly training sessions (*p* = 0.004) and, lastly, a greater volume of weekly training (*p* = 0.003) than athletes without injury. Moreover, the most important training factors that reduce the likelihood of suffering an injury were taken into account-in. particular, resistance training alone (*p* = 0.011) or together with CORE training (*p* = 0.006) or stretching (*p* = 0.012), and the dominant side of paddling (*p* = 0.032). In conclusion, resistance training would seem to reduce the likelihood of injury among SUP practitioners, and such benefits could be obtained by resistance training alone or in combination with CORE training or stretching.

## 1. Introduction

Stand-up paddleboarding (SUP) is an increasingly popular water sport, born from surfing with modern roots in Hawaii [1]. SUP paddleboarders stand on boards that float on the water and use a paddle to propel themselves through the water. This type of sport requires good balance and great strength in the trunk muscles [2], as well as a well-developed aerobic capacity [3]. However, its practice (like other water sports) is not devoid of the risk of injury [4,5]. In this regard, the most common pathologies affecting SUP paddlers tend to be those linked to the shoulders, lower back and wrist [6,7]. In particular, 31–32.9% of paddlers had reported shoulder pain [6,7], while 25–33% also had back pain [8,9]. For that reason, it has been suggested that risk of injury may be related to a number of factors including: uneven surface conditions [8,9], poor technique with unusual movement patterns and force-profile asymmetry [10,11,12,13], the hours spent performing SUP [6] and the repetitive nature of paddling [14,15]. In addition, other factors previously identified in other types of sport might also influence the likelihood of injury to the SUP paddle boarder. Some of these variables are: stretching [16,17,18] or core strengthening [19,20,21] and resistance training (RT) [22,23,24,25].

There have been studies conducted on the lack of flexibility regarding the greater likelihood of injury in hamstrings [26]. However, it might be the stretching technique that increases flexibility beyond that required for sport-specific movements which causes these types of injury [17]. Given the different demands and ranges of motion, the vast majority of studies do not most likely differentiate between different types of injury, and so it is difficult to ascertain the relationship between flexibility and the likelihood of injury [18]. However, as far as CORE training is concerned, it would seem that literature is more in agreement, since following 8 weeks of CORE training, improvements in postural control and quality of movement have been reported in university athletes [19], or improvements in reaction to jumps in female athletes following 6 weeks of CORE training [20], which could have a real impact on injury prevention, although studies are still required to further clarify this statement.

With regard to RT, it has been shown that it is the only modifiable risk factor that contributes significantly to the likelihood of suffering a sports injury [27]. It has been known for years that RT can help prevent injuries due to overuse such as swimmers’ shoulders or tennis elbows [28]. In this regard, RT would appear to have a direct relationship with the probability of injury in active people, it being demonstrated that 4 weeks of resistance training reduced the possibility of injury in hamstrings [29] and, in the case of the military, a 14-week program with 4 daily resistance training exercises involving concentric and functional eccentric contractions of the hip and knee extensors was applied to prevent anterior knee pain in military recruits, with positive effects being obtained [30]. In types of sport such as soccer, it was demonstrated how 12 weeks of RT significantly reduced the number of injuries in young players [31], or in running, for which resistance training is recommended to prevent future injuries [32], enabling the athlete to perform successfully [33]. It should be noted that the effect of this type of training is not only valid in reducing the likelihood of injury, but also in reducing its severity. To this end, a study was carried out on high school athletes, in which it was shown that athletes who had undertaken RT suffered fewer injuries (26.2% vs. 72.4%), and also less time was lost over the injury rehabilitation period (2.02 days vs. 4.82 days) [34]. Unfortunately, no previous evidence has been reported about the influence of training factors on injuries suffered in SUP, to the best of the author’s knowledge.

For that reason, the main aim of this study focused on identifying any training factors that could influence injuries suffered by participants in international SUP competitions. We conducted an online survey, and then related this data to different variables associated with training, especially RT. The information gathered in the survey would go on to form the basis for prevention strategies in SUP sports injuries, based on previous references [6,7].

## 2. Methods

### 2.1. Participants

The research was conducted in the form of a retrospective observational cohort study. All athletes who competed in international race circuit races (Eurotour) were offered the chance to participate in this online test (2019). One hundred and six participants started the survey, nine of whom did not answer the injury section, leaving 97 participants (77.3% male and 22.7% female) that were finally included in the injury analysis (38.06 ± 11.72 years). Participation was worldwide, with the majority from Spain (69.2%), followed by the rest of Europe (24.5%), United States (4.2%), Africa and Australia (2.1%).

Ethics were approved by the University of Deusto Research Ethics Committee (ref: ETK-13/18-19), designed in accordance with the Declaration of Helsinki [35,36]. Participants received all the information detailing the study aims in advance. Their rights were preserved, with voluntary participation being requested, and the chance to withdraw at any moment being provided. Information regarding the purpose, procedures and confidentiality of the study was provided, and informed consent obtained from all participants.

### 2.2. Experimental Trials

The survey consisted of four sections: (I) introduction and informed consent, (II) demographics and participation, (III) training and competition, and (IV) injuries. Section I provided background information on the purpose of the study and electronic informed consent, with participants being unable to access the survey unless they provided such consent. Section II involved demographic and SUP participation questions, namely questions regarding age, sex, height, and body mass. Section III referred to questions related to training routines (contents of the training: flexibility, resistance, and strength), and the amount performed by participants in the course of such training). Section IV involved questions about SUP injuries and their epidemiology. The participants were instructed to provide information specifically regarding SUP injuries suffered over the last year, and in an attempt to gather data about multiple injuries on the same part of the body, they were able to provide information about more than one injury to the same part.

### 2.3. Procedure

An online Ad-hoc specific survey was specifically used to determine the epidemiology of injuries suffered by participants in international SUP competitions, via an online survey, with this data being linked to different variables attached to training, especially RT and free-text responses. The authors designed the first draft of the survey, and a pilot test was performed by two members not included in the study who volunteered to participate. This survey had been used in previously published surveys on injuries and [6,7] was available in both Spanish (Appendix A) and English (Appendix A) from 20 April 2019 to 15 July 2019, when the participants gave their responses. It comprises the six sections described in the following paragraph and took a maximum time of 15 min to be completed. An email was sent to the participants of an international SUP race circuit, in which they were informed of the objectives set out by the study and asked to participate. The questionnaire was sent out twice and participants were allowed to respond to it over a period of three months.

### 2.4. Statistical Analysis

The results were shown as frequencies and % of cases. Kolmogorov–Smirnov tests were carried out to test the normality of the studied continuous variables (*n* > 50). and the Levene test was used to check the homoscedasticity of the differences in variance of different descriptive and/or training data between male and female, assessed via a one-way ANOVA test with the injury as a mixed factor. Moreover, the multi-factor ANOVA test was used to identify those factors which influence the state of injury. For their part, effect sizes were calculated using partial square eta (η^2^p), and interpreted according to the one indicating that there is no effect if 0 ≤ η^2^p < 0.05; minimal effect if 0.05 ≤ η^2^p < 0.26; moderate effect if 0.26 ≤ η^2^p < 0.64; and a major effect if η^2^p ≥ 0.64 [37]. Post hoc statistical power was calculated for paired T-test, while statistical. analysis was completed using SPSS Statistics version 24.0 (SPSS: An IBM Company, IBM Corporation, Armonk, NY, USA). Lastly, statistical significance was designated in cases where *p* < 0.05.

## 3. Results

The 97 valid responses received were distributed among participants who had suffered some kind of injury and those who had not suffered any injury when practicing SUP. Demographic and body composition data were related to the likelihood of suffering an injury, and both the number of years’ practice of the activity and the number of annual or international competitions performed were positively related (Table 1).

In terms of the epidemiology of recorded lesions (Table 2), almost 60% occurred in the arms or in the upper thoracic region, revealing which parts of the body are most vulnerable during SUP. Regarding diagnosis, approximately 65% of injuries were in tendons or muscles and almost half of cases were related to overuse.

Of all the variables consulted related to training, four were significantly different depending on injured or non-injured groups of participants (Table 3): resistance training sessions per week, months per year spent resistance training, number of weekly training sessions and volume of weekly training. Both the volume of training performed weekly (expressed in sessions or total volume) and the amount of resistance training performed (expressed in weekly sessions or months per year conducted) would appear to be the most important variables to take into consideration in order to prevent injuries.

Related to the above, the following (Table 4) shows the differences in various variables among participants who suffered an injury and those who had not, although variables such as gender or practicing a sport other than SUP would not seem to affect the likelihood of suffering an injury. RT (alone or together with CORE training or stretching) or the dominant side of paddling are the variable that gave rise to differences in the likelihood of suffering an injury while practicing SUP (*p* < 0.05).

## 4. Discussion

The main aim of the present study was to establish the training factors which influence injuries in international SUP competitions. The main finding of this research showed that RT, regardless of the number of weekly sessions, volume of training or the dominant side of paddling, was seen to have a significant relationship with the likelihood of injury in SUP practitioners.

To the best of our knowledge, scarce scientific evidence related to injury has been published in the scientific literature about SUP. As for the type of injury and recorded injuries described in our study, these do not vary much from the data obtained by Furness et al. [6]. In our case, 59% of recorded injuries were in the upper thoracic region (34.4%) or in the arms (24.6%), while in the data recorded by Furness et al. these areas were also the most affected with 44.5%. In terms of the type of injury, in 52.5% of recorded data this occurred in the tendon or muscle, while in the case of Furness et al. [6] it was 50.4% with similar percentages.

As far as flexibility training is concerned, no significant relationship was found in terms of a reduction in injuries. The literature on the subject is very confusing regarding this hot topic, given that some studies support the idea that flexibility work can reduce the likelihood of injury [38,39], or a limitation in it can increase the risk of injury [40]. However, it depends on the type of training system used, type of sport practiced or other variables [18]. Conversely, in some studies, the decrease in flexibility is associated with greater economy in terms of running [41]. Other scientific articles claim that an increase in flexibility beyond that necessary for the type of sport involved can cause injuries [17]…and so no conclusive statements can be made about the relationship between flexibility and the likelihood of injury [18]; the fact that no relationship was found may be due to the type of flexibility training performed by each athlete and the different types of injury recorded. Therefore, further research is necessary in this field.

The literature is scarce when it comes to CORE training and reduction in the likelihood of injury. Improvements have been described following 8 weeks of CORE training in movement patterns with college athletes [19], in the jumping pattern of dancers after 6 weeks of CORE training [20] and in firefighters, with a 42% reduction in CORE training injuries [21]. This difference between the data obtained in the study and existing literature may be due to the fact that SUP is a sport that is very demanding in terms of balance [2]. Therefore, it is assumed that SUP practitioners will make a major effort to stabilize muscles, including CORE. For that reason, specific work on this may not give rise to improvements.

The fact that those who are least injured train almost twice as many months a year in RT and undertake almost twice as many weekly RT sessions as those who are most injured highlights the importance not only of RT, but of training in it for a sufficient of time and with a sufficient weekly load. In this regard, there are studies [42] that show that a preventive program of only 10–15 min’ duration is enough to achieve a 45% reduction in the likelihood of injury, although this will of course also depend on different factors such as: the characteristics of the participant; the type of injury training program carried out or other types of variable. There is a trend towards increasingly injuries being reduced in those studies with a longer intervention phase compared to others with shorter intervention periods [43,44,45,46]. There are studies that have recorded improvements in strength with intervention programs of only 2–4 weeks’ duration [47,48,49], with this increase most likely caused by neuromuscular and connective tissue adaptations [50], rather than an increase in muscle. As for the reduction in injuries, improvements have been reported in intervention studies of only 4 weeks’ duration [51,52]. Having said this, it is likely that the participants interviewed underwent too little training load to be sufficiently adapted, although we do not know what type of work is the most suitable.

It should be noted that the vast majority of research conducted so far, in which an injury prevention program is applied so as to research into its effect on a population, does not distinguish between different types of injury and types of RT [53]. This is the main reason why it is difficult to associate one type of RT with the prevention of a particular injury. While it is clear that RT plays an important role in injury prevention and rehabilitation [54], it is also clear that there is no single optimal RT program for all sports yet to our knowledge. Thus, an appropriate training program should take into account the following variables: characteristics of the participants, the main aims of the program, and the type of injury to be prevented, as well as the muscle imbalances between agonists and antagonists [55]. Some researchers stress the importance of evaluating muscle imbalances between agonist-antagonists, as well as the same muscle groups at different extremities, with the aim of detecting athletes with a greater predisposition to injury [28].

It would seem that RT improves the capacity of the muscles to work for long periods of time, as well as increasing the elasticity of the tendon-aponeurosis structures [56], given that heavy loads lead to greater neuromuscular improvements. In the case of programs aimed at strengthening connective tissue, such as ligaments, eccentric activations would seem to be the most appropriate [57] because they generate more tension with less metabolic stress [58]. However, this type of training would not seem to have the same potential for the tendons, given that the collagen metabolism appears not to be affected by such activations [57].

In muscles, a reduction in muscle mass is a factor that increases the likelihood of injury and RT is an effective way of avoiding this [59]. Adaptations in the muscle occur at different stages, and at the beginning of the program there are rapid improvements in strength due to neuromuscular adaptations, followed by a slow progression as the muscle increases its cross-sectional area [60,61]. The neuromuscular adaptations observed are mainly: acquisition of a motor function by the nervous system, increased muscle activation and improved synchronization of motor units and improved intramuscular coordination [62]. In terms of hypertrophy, the main adaptation involves an increase in the cross-sectional area of the fibers, as well as an increase in the number of sarcomeres [63].

In addition to these adaptations, bone adaptations also occur as a result of RT training, mainly improvements in bone density and, therefore, in bone strength, and it would seem that RT training is one of the most osteogenic effects [59]. In the case of connective tissue, however, adaptations would seem to occur in both the increase in size and strength of these tissues [54,57], the increase in size appearing to be the result of the increase in collagen, with the latter being proportional to the increase in muscle. Thus, everything points to the fact that the increase in muscle mass corresponds to the increase in size and strength of the connective tissue [59]. In addition, researchers have shown that injured tendons and ligaments recover faster when athletes undertake RT [56]. The influence of genetics and nutrients on RT adaptations in each individual [64,65,66,67], based on anatomical variability, should therefore not be overlooked.

However, the risks attached to RT must be considered, as well as the risk of training with loads or inappropriate volumes for strength training, although most scientific literature indicates that RT is safer than many other types of activity, especially when performed under supervision [34]. Some studies in fact point out the danger of RT [68], insofar as performing. volumes and intensities greater than the subject can assimilate could increase the risk of injury [69]. In addition, RT with heavy loads prior to some activities could be harmful and increase the risk of injury due to the fatigue it leaves in the tissues [70]. This only reinforces ideas about the importance of personalized training and proper application between load and recovery [71], as well as the gradual conditioning of the tissues [30].

On the other hand, although RT improvements in performance [72] and health [73] are well known, the effect and mechanisms of RT on injury prevention have not yet been well documented [55]. Having said this, it is impossible to avoid injuries completely, although there would seem to be ways of reducing the risk and severity of injuries by progressively increasing the tensile strength of the tissue [55]. However, the results obtained from this study failed to find any improvements in CORE and flexibility training, bearing in mind the existing confusing literature and the need for more studies that delve deeper into the different training systems of each content as well as the different injuries. Therefore, we cannot recommend training in them, especially given their low risk and the low volume or workload needed to obtain improvements. For this reason, a multi-component prevention training routine is advocated which would seem to be the most suitable way of preventing both the amount and variety of injuries [74], albeit always with RT as the main component.

Lastly, it is important to take into account the importance of dominance of one side or the other when paddling in view of the likelihood of suffering an injury. As in the case of other types of such activity, there are studies that defend the idea that asymmetries increase the likelihood of suffering an injury [75], or even decompensation deriving from unilateral daily activities [76]. In the data reported by paddlers, the dominance of one side of the paddle would also seem to be linked to the likelihood of suffering an injury. This may be due to the fact that when practicing SUP, even if paddlers are paddling on both sides of the board alternately [2], experienced paddlers, such as those subject to study here, are able to paddle by keeping the board in the right direction. Thus, the main load is supported by the paddler’s dominant side.

### Limitations and Strength and Future Lines of Research

Some limitations in the study should be recognized. The main one is the free interpretation that participants need to make in defining the concepts of flexibility, core and resistance training. All concepts encompass different training systems and while some may be beneficial, others may have no effect or may even have a negative effect on the likelihood of injury. On the other hand, the study sample is small, despite considering the type of sport that is SUP and the number of participants who usually take part in this type of competition. Besides, the sample has a greater significance. Based on this first approach, we can recommend training program strategies in order to reduce the injury rate when athletes practice or compete in SUP.

Further studies, like the present one, are needed in other countries, as educational backgrounds differ from country to country and athletes’ preferences may vary according to their gender, age or ethnicity and culture, with different possibilities replicating the methodology used in the present study arising. The use of cross-sectional vs. longitudinal surveys may offer the potential for further chances for scrutiny in parallel future studies. However, this approach may miss some relevant information regarding the motivation behind coaches and practitioners.

## 5. Conclusions

In summary, resistance training, alone, with CORE or stretching, has been shown to play a key role in the prevention of injuries in SUP among athletes who have competed in international circuit races. Thus, unlike CORE and flexibility training, resistance training reduces the likelihood of injury among SUP competitors. Moreover, the number of weekly sessions, as well as the volume of training, would seem to be related to injury in SUP international athletes. On the other hand, over 50% of registered injuries are located in the arms or in the upper thoracic region.

### Practical Applications

In this study we tried to ascertain the relationship between flexibility training, CORE and RT with the likelihood of injury in SUP. RT has tended to be associated with a reduction in recorded injuries, although this has not been the case with CORE or flexibility training. This concept may help coaches and team physician members to improve the design of their athletes’ routines, and hence improve continuity in training and performance.

## Figures and Tables

**Table 1 ijerph-18-00880-t001:** Demographic and Body composition data of the participants based on state of injury.

	No (*n* = 36)	Yes (*n* = 61)	*p*	η^2^p	Post Hoc Power
Age (Year)	38.31 ± 13.29	37.95 ± 11.09	0.897	0.000	0.050
Body mass (Kg)	75.77 ± 14.60	72.85 ± 12.72	0.352	0.010	0.135
Height (m)	175.36 ± 10.52	174.59 ± 7.71	0.679	0.002	0.070
Body Mass Index	24.30 ± 3.78	23.79 ± 3.08	0.471	0.005	0.110
Practice of sup (year)	1.54 ± 0.65	1.82 ± 0.53	0.038	0.050	0.551
Competitions per year	2.54 ± 0.95	3.30 ± 0.97	0.001	0.116	0.957
International competitions	1.92 ± 0.98	2.59 ± 0.74	0.001	0.125	0.969

Data expressed as mean ± standard deviation. *p*-value: significant differences according to state of injury by one ANOVA factor. No: Has not suffered any injury, Yes: Has suffered some injury.

**Table 2 ijerph-18-00880-t002:** Characteristics of recorded injuries.

		Total (*n* = 61)	
		*n*	%
**Anatomical Area**	Head	2	3.3
	Arm	15	24.6
	Upper thoracic region	21	34.4
	Back	11	18.0
	Lower Body	12	19.7
**Diagnosis**	Tendinitis	21	34.4
	Irritation	2	3.3
	Subluxation or sprain	4	6.6
	Concussion	2	3.3
	Fracture-muscle damage	11	18.0
	Luxation	2	3.3
	Muscle Contracture	8	13.1
	Superficial Wound	3	4.9
	Others	8	13.1
**Type of Injury**	New injury	50	82.0
	Relapse of Injury	11	18.0

**Table 3 ijerph-18-00880-t003:** Training characteristics of participants based on state of injury.

	No (*n* = 36)	Yes (*n* = 61)	*p*	η^2^p	Post Hoc Power
Resistance sessions per week (day)	1.85 ± 1.46	1.11 ± 1.37	0.028	0.056	0.600
Months per year of resistance training	8.23 ± 4.68	4.11 ± 4.89	<0.001	0.135	0.950
Training amount (days per week)	3.00 ± 1.39	4.07 ± 1.60	0.004	0.093	0.831
Training sessions per day	1.15 ± 0.46	1.28 ± 0.49	0.271	0.014	0.195
Average session duration (hour)	1.42 ± 0.39	1.59 ± 0.39	0.073	0.037	0.435
Weekly training volume (hour)	4.81 ± 2.87	8.62 ± 5.96	0.003	0.102	0.869
Maximum session volume (hour)	2.96 ± 1.22	3.51 ± 1.64	0.130	0.027	0.327
Training for another sport (days per week)	2.55 ± 1.30	2.76 ± 1.14	0.488	0.007	0.106
Core training (days per week)	2.40 ± 1.54	2.53 ± 1.50	0.763	0.002	0.060
Flexibility training (days per week)	2.65 ± 1.77	2.86 ± 1.66	0.623	0.003	0.078
Resistance training + core (days per week)	4.00 ± 2.34	3.76 ± 2.22	0.681	0.003	0.069
Resistance training + core + flexibility (days per week)	8.28 ± 4.52	7.58 ± 3.67	0.461	0.007	0.113
Resistance training + flexibility (days per week)	4.25 ± 2.52	3.96 ± 2.19	0.611	0.003	0.080

Data expressed as mean ± standard deviation. *p*-value: significant differences according to state of injury by one ANOVA factor. No: Has not suffered any injury, Yes: Has suffered some injury.

**Table 4 ijerph-18-00880-t004:** Injured participant’s characteristics.

		No	Yes	*p*	η^2^p	Post Hoc Power
**Gender**	Male	24.1%	51.7%	0.547	0.004	0.092
	Female	5.7%	18.4%			
**Dominant side**	Right	9.1%	37.9%	0.032	0.054	0.577
	Left	11.5%	21.8%			
	Both	9.2%	10.3%			
**Practicing another Sport**	Yes	25.3%	50.6%	0.250	0.016	0.209
	No	4.6%	19.5%			
**Resistance training**	Yes	23%	32.2%	0.011	0.075	0.735
	No	6.9%	37.9%			
**Core training**	Yes	23%	39.1%	0.081	0.036	0.415
	No	6.9%	31.0%			
**Stretching**	Yes	14.9%	32.2%	0.723	0.001	0.064
	No	3.4%	11.5%			
	Sometimes	11.5%	26.4%			
**Other sport + Resistance training**	Yes	2.3%	4.6%	0.814	0.001	0.056
	No	27.6%	65.5%			
**Other sport + Resistance training + Stretching**	Yes	2.3%	3.4%	0.584	0.004	0.084
	No	27.6%	66.7%			
**Other sport + Resistance training + Stretching + Core**	Yes	2.3%	2.3%	0.351	0.010	0.153
	No	27.6%	68.8%			
**Other sport + Resistance training + Core**	Yes	2.3%	3.4%	0.584	0.004	0.084
	No	27.6%	66.7%			
**Resistance training + Stretching**	Yes	21.8%	27.6%	0.006	0.088	0.804
	No	8.0%	42.5%			
**Resistance training + Core**	Yes	19.5%	24.1%	0.012	0.074	0.723
	No	10.3%	46.0%			
**Cooling Down**	Yes	13.8%	34.5%	0.657	0.02	0.073
	No	4.6%	16.1%			
	Sometimes	10.3%	19.5%			

*p*-value: significant differences between injured state by multifactorial ANOVA. No: Has not suffered any injury, Yes: Has suffered some injury.

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
