# Peer review of "Relationship between Training Factors and Injuries in Stand-Up Paddleboarding Athletes"

_ijerph, 2021, doi:10.3390/ijerph18030880_

Round 1
Reviewer 1 Report
This is an interesting paper which is ambitious in it's data collection, seeking questionnaire data from many countries, on a topic with many variables for conclusive research claims. I commend the authors/researchers but would invite them to consider their conclusion and also some English phrasing in the second half of the paper needs attention - short sentences which fail to make a point, or full stops used where they should be commas, all hinder the reader. In sum, greater clarity of claims being made from this project to be set out in the conclusion, which is currently only 4 lines long. Attend to phrasing throughout, punctuation and reducing, finding greater efficiency in language to impart your message.
Author Response
Point-by-Point Response to Reviewer’s Comments
We would like to sincerely thank the reviewers for their helpful recommendations. We have seriously considered all the comments and carefully revised the manuscript accordingly. Revisions are highlighted in yellow through the manuscript to indicate where changes have taken place. We feel that the quality of the manuscript has been significantly improved with these modifications and improvements based on the reviewers’ suggestions and comments. We hope our revision will lead to an acceptance of our manuscript for publication in International Journal of Environmental Research and Public Health.
In advance,
King regards
REVIEWER
This is an interesting paper which is ambitious in it's data collection, seeking questionnaire data from many countries, on a topic with many variables for conclusive research claims. I commend the authors/researchers but would invite them to consider their conclusion and also some English phrasing in the second half of the paper needs attention - short sentences which fail to make a point, or full stops used where they should be commas, all hinder the reader. In sum, greater clarity of claims being made from this project to be set out in the conclusion, which is currently only 4 lines long. Attend to phrasing throughout, punctuation and reducing, finding greater efficiency in language to impart your message.
AUTHORS: Thank you for the observations.
- The text has been revised and incorrect marks have been corrected.
- The conclusion has been revised. The new conclusion is: “In conclusion, resistance training, alone, with CORE or stretching, has been shown to play a key role in the prevention of injuries in SUP among athletes who have competed in international circuit races. Thus, unlike CORE and flexibility training, resistance training reduces the likelihood of injury among SUP competitors. Moreover, the number of weekly sessions, as well as the volume of training, would seem to be related to injury in SUP international athletes. On the other hand, over 50% of registered injuries are located in the arms or in the upper thoracic region.”
- The text has been revised by removing unnecessary sentences.

Reviewer 2 Report
Congratulations to the authors. This is an original study and the results are very interesting. The objective of the paper was identify the training factors that could influence on injuries suffered by participants in international competitions of SUP. Also, the authors deal with an interesting and important topic from a scientific and practical point of view, and the study is well designed. In this line, there are only minor comments that should be addressed:
I present the following in the order which the paper was written:
Introduction
- Line 41 and 42: There are some words that have different font size.
- Line 79: Is there any type of study that has studied this topic in other water sports such as surfing, windsurfing, etc?
Materials and Methods
- Line 118: Was the survey only available in Spanish?
- Methods are well described. Did you perform some type of statistical power calculation; at least a-posteriori?
Results
- It could be interesting to indicate in the legend of the table the meaning of Yes and No.
Discussion
- The discussion is very complete and is well oriented to explain the research problem.
- Line 170: It seems that there is an error in the citation format and the capital letter should be removed.
- Line 175: References should be separated by commas.
- Line 193: Please, change “results. being another...” by “results, being another...”.
- Line 210: “It” should be in lower case.
- Line 219: It seems that this is a redundancy: “muscle resistance RT…”. Please amend it.
- Line 222 and 223: Please change the dots to commas. Review it throughout the document.
Conclusion
- It would be interesting to include some limitations of the study and practical applications.
Author Response
Point-by-Point Response to Reviewer’s Comments
We would like to sincerely thank the reviewers for their helpful recommendations. We have seriously considered all the comments and carefully revised the manuscript accordingly. Revisions are highlighted in yellow through the manuscript to indicate where changes have taken place. We feel that the quality of the manuscript has been significantly improved with these modifications and improvements based on the reviewers’ suggestions and comments. We hope our revision will lead to an acceptance of our manuscript for publication in International Journal of Environmental Research and Public Health.
In advance,
King regards
REVIEWER
Congratulations to the authors. This is an original study and the results are very interesting. The objective of the paper was identify the training factors that could influence on injuries suffered by participants in international competitions of SUP. Also, the authors deal with an interesting and important topic from a scientific and practical point of view, and the study is well designed. In this line, there are only minor comments that should be addressed:
I present the following in the order which the paper was written:
Introduction
REVIEWER: Line 41 and 42: There are some words that have different font size.
AUTHORS: Thank you for your observation, the authors have corrected these differences.
REVIEWER: - Line 79: Is there any type of study that has studied this topic in other water sports such as surfing, windsurfing, etc?
AUTHORS: Thanks for the interest. In the scientific literature there are some epidemiological studies that show injuries in some water sports such as surfing. These studies are referenced on line 44-45. However, to the knowledge of the authors, there are no studies that show the influence of training factors on SUP injuries. Therefore, the authors have rewritten the following paragraph in the text: “Unfortunately, no previous evidence has been reported about the influence of training factors on injuries suffered in SUP, to the best of the author’s knowledge.”
Materials and Methods
REVIEWER: Line 118: Was the survey only available in Spanish?
REVIEWER - Methods are well described. Did you perform some type of statistical power calculation; at least a-posteriori?
AUTHORS: Thanks for your comment. The authors have included this information in table 1 and table 2.
Results
REVIEWER: It could be interesting to indicate in the legend of the table the meaning of Yes and No.
AUTHORS: Thanks for the observation. The meaning of No and Yes has been added in the legends of the tables.
Discussion
The discussion is very complete and is well oriented to explain the research problem.
REVIEWER: Line 170: It seems that there is an error in the citation format and the capital letter should be removed.
AUTHORS: Thank you for your interest. The error has been corrected.
REVIEWER: Line 175: References should be separated by commas.
AUTHORS: Thank you for your observation. Mistakes have been corrected.
REVIEWER: Line 193: Please, change “results. being another...” by “results, being another...”.
AUTHORS: Thank you for your recommendation. The sentence has been deleted because both the limitations and the future lines of research are discussed in another paragraph of the article.
REVIEWER: Line 210: “It” should be in lower case.
AUTHORS: Thank you for your recommendation. The error has been corrected.
REVIEWER: Line 219: It seems that this is a redundancy: “muscle resistance RT…”. Please amend it.
AUTHORS: Thank you for your recommendation. The error has been corrected.
REVIEWER: Line 222 and 223: Please change the dots to commas. Review it throughout the document.
AUTHORS: Thank you for your recommendation. The mistakes have been corrected and the text revised.
Conclusion
REVIEWER: It would be interesting to include some limitations of the study and practical applications.
AUTHORS: Thank you for your recommendation. Limitations and strength and future research lines are developed in the final part of the discussion, instead, practical applications are developed in the final part of the conclusion.

Reviewer 3 Report
Dear Authors,
Thank you very much for your interesting research. The contents of your article is of high relevance to the community of sports sciences and sports medicine and clearly worth being published. However, your manuscript in its current form is subject several flaws that need to be addressed carefully and thoroughly before I can recommend its publication
Major issues:
- Especially lines 19-84 and 126–135, but also throughout manuscript: Poor English, hard to understand, sometimes seemingly self-contradicting, probably because of vocabulary, punctuation and/or grammar errors
- For instance: Line: 29–33: “Resistance training sessions per week were(p=0.028), months per year that resistance is trained(p=0.001), […] were the variables positively related to 31 the probability of injury. Resistance training was the most important training factor decreasing the 32 probability of suffering an injury, alone (p=0.008) or together with CORE training(p=0.004)or 33 stretching(p=0.008).”
That paragraph is highly confusing and remains unclear as it contains contradictory statements: Is resistance training a factor increasing or decreasing the injury risk in SUP? The phrase “positively related to the probability of injury” mathematically states that the probability of an injury is increased by an increasing level /amount of resistance training. Maybe you mix up positive correlations (objective mathematical definition) and positive outcomes (subjective assessment of results). - Line: 118: The internet link provided does not seem to be permanent in terms of long-term accessible. Please add the contents of the sites linked as appendix to your manuscript.
- Line: 141 (Table 1): A mean body mass index of 27.9+18.8 indicates overweight and is rather surprising for trained athletes on competition level in SUP. Also, the standard deviation given is strangely high. Please discuss on that. Are there perhaps any severe outliers distorting your results? In this case, the median would be a much more adequate statistical descriptor and should be used.
- Line 159 (Table 4): A multifactorial ANOVA would be much better suited for the statistical analyses you intend. Please perform that for the main factors you have identified in your pairwise comparisons.
- Line 155–157 : I disagree that the dominate side does not affect the injury risk: See the corresponding entry in your Table 4: In case that “both sides are equally dominant”, i.e. no dominant side is present in the athlete, relative injury risk is approximant 50 %, whereas it is at roughly 200 % or 400 % in the case of left-side or right-side dominance. This is a very important finding you have to discuss! From other recent studies it is known that even in symmetric sports, asymmetries beyond a certain threshold are disadvantageous and may be harmful.
References for reference:
- https://link.springer.com/chapter/10.1007/978-981-15-2549-0_4
- https://www.sciencedirect.com/science/article/abs/pii/S0949328X19301449
- https://www.mdpi.com/1010-660X/56/12/683
Minor issues:
- Line 41: Please correct font sizes.
Best regards
Your reviewer
Author Response
Point-by-Point Response to Reviewer’s Comments
We would like to sincerely thank the reviewers for their helpful recommendations. We have seriously considered all the comments and carefully revised the manuscript accordingly. Revisions are highlighted in yellow through the manuscript to indicate where changes have taken place. We feel that the quality of the manuscript has been significantly improved with these modifications and improvements based on the reviewers’ suggestions and comments. We hope our revision will lead to an acceptance of our manuscript for publication in International Journal of Environmental Research and Public Health.
In advance,
King regards
Dear Authors,
Thank you very much for your interesting research. The contents of your article is of high relevance to the community of sports sciences and sports medicine and clearly worth being published. However, your manuscript in its current form is subject several flaws that need to be addressed carefully and thoroughly before I can recommend its publication
Major issues:
REVIEWER: Especially lines 19-84 and 126–135, but also throughout manuscript: Poor English, hard to understand, sometimes seemingly self-contradicting, probably because of vocabulary, punctuation and/or grammar errors
AUTHORS: Thanks for the observation. The text has been revised by a native in order to improve the English.
REVIEWER: For instance: Line: 29–33: “Resistance training sessions per week were(p=0.028), months per year that resistance is trained(p=0.001), […] were the variables positively related to 31 the probability of injury. Resistance training was the most important training factor decreasing the 32 probability of suffering an injury, alone (p=0.008) or together with CORE training(p=0.004) or 33 stretching(p=0.008).”
That paragraph is highly confusing and remains unclear as it contains contradictory statements: Is resistance training a factor increasing or decreasing the injury risk in SUP? The phrase “positively related to the probability of injury” mathematically states that the probability of an injury is increased by an increasing level /amount of resistance training. Maybe you mix up positive correlations (objective mathematical definition) and positive outcomes (subjective assessment of results).
AUTHORS: Thanks for your comment. The text has been modified with the aim of improving its comprehensibility.
“Likewise, the results showed that athletes with injury performed fewer resistance training sessions per week (p=0.028), over fewer months per year (p=0.001), more weekly training sessions (p=0.004) and, lastly, a greater volume of weekly training (p=0.003) than athletes without injury. Moreover, the most important training factors that reduce the likelihood of suffering an injury were taken into account - in. particular, resistance training alone (p=0.011) or together with CORE training (p=0.006) or stretching (p=0.012), and the dominant side of paddling (p=0.032).”
REVIEWER: Line: 118: The internet link provided does not seem to be permanent in terms of long-term accessible. Please add the contents of the sites linked as appendix to your manuscript.
AUTHORS: Thank you very much for your contribution. The online questionnaire, in PDF format, has been downloaded in Spanish and English and they are attached as an appendix.
REVIEWER
- Line: 141 (Table 1): A mean body mass index of 27.9+18.8 indicates overweight and is rather surprising for trained athletes on competition level in SUP. Also, the standard deviation given is strangely high. Please discuss on that. Are there perhaps any severe outliers distorting your results? In this case, the median would be a much more adequate statistical descriptor and should be used.
AUTHORS: Thank you for your observation. After reviewing the data base, we have corrected these data. The correct height and BMI is:
Height (m) |
175.36±10.52 |
174.59±7.71 |
0.679 |
0.002 |
0.070 |
Body Mass Index |
24.30±3.78 |
23.79±3.08 |
0.471 |
0.005 |
0.110 |
REVIEWER: Line 159 (Table 4): A multifactorial ANOVA would be much better suited for the statistical analyses you intend. Please perform that for the main factors you have identified in your pairwise comparisons.
AUTHORS: Thank you for your recommendation. The authors have change statistics in table 4. In this table has been included data from multifactorial ANOVA. Likewise, some phrases about this change have been added along the manuscript (abstract, material and methods results, and discussion sections).
REVIEWER
- Line 155–157: I disagree that the dominate side does not affect the injury risk: See the corresponding entry in your Table 4: In case that “both sides are equally dominant”, i.e. no dominant side is present in the athlete, relative injury risk is approximant 50 %, whereas it is at roughly 200 % or 400 % in the case of left-side or right-side dominance. This is a very important finding you have to discuss! From other recent studies it is known that even in symmetric sports, asymmetries beyond a certain threshold are disadvantageous and may be harmful.
References for reference:
- https://link.springer.com/chapter/10.1007/978-981-15-2549-0_4
- https://www.sciencedirect.com/science/article/abs/pii/S0949328X19301449
- https://www.mdpi.com/1010-660X/56/12/683
AUTHORS: Interesting contribution. A paragraph has been added to the discussion commenting on the data obtained related to the dominant side of the paddlers and the probability of suffering an injury:
“Lastly, it is important to take into account the importance of dominance of one side or the other when paddling in view of the likelihood of suffering an injury. As in the case of other types of such activity, there are studies that defend the idea that asymmetries increase the likelihood of suffering an injury [75], or even decompensation deriving from unilateral daily activities [76]. In the data reported by paddlers, the dominance of one side of the paddle would also seem to be linked to the likelihood of suffering an injury. This may be due to the fact that when practicing SUP, even if paddlers are paddling on both sides of the board alternately [2], experienced paddlers, such as those subject to study here, are able to paddle by keeping the board in the right direction. Thus, the main load is supported by the paddler’s dominant side.”
Minor issues:
REVIEWER: Line 41: Please correct font sizes.
AUTHORS: Thanks very much. The mistakes have been corrected.

Reviewer 4 Report
Attached file with data for the author.

Author Response
Point-by-Point Response to Reviewer’s Comments
We would like to sincerely thank the reviewers for their helpful recommendations. We have seriously considered all the comments and carefully revised the manuscript accordingly. Revisions are highlighted in yellow through the manuscript to indicate where changes have taken place. We feel that the quality of the manuscript has been significantly improved with these modifications and improvements based on the reviewers’ suggestions and comments. We hope our revision will lead to an acceptance of our manuscript for publication in International Journal of Environmental Research and Public Health.
In advance,
King regards
REVIEWER
OBJECT OF THE STUDY: This article analyzes the injuries suffered by Stand-up paddle (SUP) international athletes, during their practice and training, relating it to demographic and body composition variables. Data collection was carried out on 97 paddlers from the international SUP circuit. After analyzing the data and discussing the contributions of other authors, it is concluded by stating that resistance training was the training factor that most reduces the possibility of injury in the practice of this sport
REVIEWER: Regarding the title, in the statement proposed by the authors it is appropriate, using the keywords that are included in the objective of the work, in a short and concrete way.
The introduction provides sufficient background to contextualize the article regarding the studies on sports injuries and training, in other similar sports specialties. References used from other named works are relevant and up to date.
Regarding the research method, the study population is clearly identified and selected, as well as the justification for the final number of participants and their origin. One piece of information that does not appear in this section and it would be appropriate to comment is the age of the participants, even if it is recorded in the results.
The research design and measurement procedure are adequate and well explained in the text. The tool used, although it went through a pilot test, would be more reliable if it had been validated by a group of specialists.
The field work and data collection has been carried out with the approval of the research ethics committee of the University of Deusto, referenced in the text.
In the results section the statistics are adequate, although the structure of the tables can be improved on a graphic level, taking the title out of the table bars and differentiating the table number and its title on different lines.
Regarding the discussion, it is correctly argued and the different variables studied are related, with similar and also different conclusions from other authors. The 2 references used in the discussion are appropriate and in line with the research. In this section, specifically between lines 189-193, one of the limitations of the research is advanced, which should not be mentioned in this section, since it is done in section 4.1 of limitations and future lines of study.
The conclusions are a direct consequence of the objective of the study and the very clear and significant practical applications.
AUTHORS:
Thank you for your comments and contributions.
- The age of the participants has been added in the methods. Line 92. (38.06±11.72 years)
- The structure of tables has been modified, considering the title of the table out, the table bars and differentiating the table number and its title on different lines.
- The sentence referring to the limitation (line 191-195) has been deleted.
REVIEWER
STRENGTHS OF THE ARTICLE
- New vision of SUP training work, which provides concrete results to apply in sports redemption.
- Novelty in relation to a very current and little known sports specialty and with little scientific research behind
- Good structuring of the article.
- Current references on injuries and sports training of the SUP specialty
- Clarity in the results
- Good conclusions.
ARTICLE WEAKNESSES
- Decompensation in the number of participants by sex.
- The tool could be more reliable
- The countries of origin of the participants may condition the results obtained.
The English language and style are correct with good spelling, although there are some deficiencies in the writing style but of a minor nature.
In conclusion, after reviewing the content of the article and studying the design and method of the research, the structuring of the text, the scientific arguments provided and the conclusions reached by the authors, my recommendation to the editor is to accept after a minor revision (minor bug fixes regarding the form and the structure of the results tables).
AUTHORS:
Thanks for the observations.
- Although we would have liked to have obtained a more equal sample in terms of gender, participation in this type of test is still very unequal, and it is difficult to get women interested. Women's participation in this type of test is usually less than 30% comparing with male, https://www.eurotoursup.net/copy-of-2019-results . In our study, we have obtained the 22.7% of female participation, which we believe is not very far from the reality of this type of sport.
- Thank you for your interest. This survey has been used in previously published studies on injuries [4,5].
- Although it was valued to analyse the sample according to the country of origin, we obtained few responses from many different countries (except Spain which was the country with the largest number of participants), and therefore, the results would not have enough statistical power. This is the main reason why we decided to make a classification based on continent.

Round 2
Reviewer 3 Report
Dear Authors,
Thank you for your careful and thorough revision. I am now happy to recommend publication of your interesting article in IJERPH.
Best regards
Your reviewer